# Serum Concentration of Growth Differentiation Factor 15 and Atherosclerosis among General Older Japanese Individuals with Normal Weight

**DOI:** 10.3390/biomedicines11061572

**Published:** 2023-05-29

**Authors:** Yuji Shimizu, Naomi Hayashida, Hirotomo Yamanashi, Yuko Noguchi, Shin-Ya Kawashiri, Midori Takada, Kazuhiko Arima, Seiko Nakamichi, Yasuhiro Nagata, Takahiro Maeda

**Affiliations:** 1Department of General Medicine, Nagasaki University Graduate School of Biomedical Sciences, Nagasaki 852-8523, Japan; yamanashi@nagasaki-u.ac.jp (H.Y.); tmaeda@nagasaki-u.ac.jp (T.M.); 2Epidemiology Section, Division of Public Health, Osaka Institute of Public Health, Osaka 537-0025, Japan; takada@iph.osaka.jp; 3Division of Strategic Collaborative Research, Atomic Bomb Disease Institute, Nagasaki University, Nagasaki 852-8523, Japan; naomin@nagasaki-u.ac.jp; 4Leading Medical Research Core Unit, Nagasaki University Graduate School of Biomedical Sciences, Nagasaki 853-8523, Japan; ynagata1961@nagasaki-u.ac.jp; 5Department of Community Medicine, Nagasaki University Graduate School of Biomedical Sciences, Nagasaki 852-8523, Japan; y-noguti@nagasaki-u.ac.jp (Y.N.); shin-ya@nagasaki-u.ac.jp (S.-Y.K.); 6Department of Public Health, Nagasaki University Graduate School of Biomedical Sciences, Nagasaki 852-8523, Japan; arima@nagasaki-u.ac.jp; 7Nagasaki University Health Center, Nagasaki 852-8521, Japan; seiko-n@nagasaki-u.ac.jp

**Keywords:** atherosclerosis, CIMT, GDF-15, normal weight, older, mitochondria, energy

## Abstract

Growth differentiation factor 15 (GDF-15), which modulates cellular energy balance, is reported to be positively associated with cardiovascular disease. However, there have been no reports about the association between serum GDF-15 concentration and atherosclerosis as evaluated by carotid intima-media thickness (CIMT) among the general population. A cross-sectional study of 536 Japanese individuals aged 60 to 69 years was conducted. To avoid the influence of abnormal cellular energy balance, this study only included participants who had a normal body mass index (BMI) and normal thyroid hormone (free thyroxine and free triiodothyronine) levels. A significant positive association between serum GDF-15 concentration and atherosclerosis was observed. In the sex- and age-adjusted model (Model 1), the odds ratio (OR) (95% confidence interval (CI)) for the logarithmic value of GDF-15 and atherosclerosis was 2.62 (1.67, 5.87). This association remained after adjusting for thyroid function and renal function (Model 2) and further adjusting for known cardiovascular risk factors (Model 3). The corresponding values were 2.61 (1.15, 5.93) for Model 2 and 2.49 (1.08, 5.71) for Model 3, respectively. Serum GDF-15 concentrations could help us to estimate the risk of atherosclerosis by indicating the status of cellular energy balance, which is related to mitochondrial activity among comparative healthy older individuals.

## 1. Introduction

Mitochondrial dysfunction is widely implicated in aging and diseases related to aging via the dysfunction of metabolic pathways and free radical production [1]. The main function of the mitochondria is the production of adenosine triphosphate (ATP) for cellular energy needs. During mitochondrial stress and dysfunction, the production of growth differentiation factor 15 (GDF-15), a stress-response member of the transforming growth factor-β (TGF-β) cytokine superfamily, is stimulated [2].

GDF-15 levels could indicate the status of cellular energy balance [3].

Although GDF-15 protects against aging-mediated systemic inflammatory responses [4], GDF-15 concentrations are reported to be positively associated with cardiovascular disease [5] and chronic kidney disease (CKD) [6].

The development of atherosclerosis, as evaluated by carotid intima-media thickness (CIMT), is an established cardiovascular risk factor [7]. It is also reported to be associated with CKD [8]. Serum concentrations of GDF-15 may be positively associated with atherosclerosis as evaluated by CIMT.

Previous studies with patients on maintenance hemodialysis [9] and patients with transfusion-dependent beta-thalassemia [10], rheumatoid arthritis [11], or psoriasis [12] have reported positive associations between CIMT and serum GDF-15 concentration. However, there have been no reports about the association between atherosclerosis as evaluated by CIMT and serum GDF-15 concentration in a comparative healthy general population.

An experimental study with a mouse model reported the paradoxical effect of GDF-15 on inflammation. GDF-15 promotes indirect proinflammatory effects in models of atherosclerosis [13,14], while GDF-15 mediates anti-inflammatory effects in mice with acute myocardial infarction [15].

Since low-grade inflammation is a known risk factor for atherosclerosis and CKD [16], clarifying the association between serum GDF-15 concentrations and atherosclerosis as evaluated by CIMT in a comparatively healthy population can help estimate the risk for developing age-related atherosclerosis.

CIMT values are significantly associated with age [17], and GDF-15 is a potential biomarker of aging [18]. Therefore, the association between GDF-15 and atherosclerosis could be strongly confounded by age. Furthermore, GDF-15 is a stress-responsive cytokine that can modulate energy balance [19]. Therefore, an abnormal body mass index (BMI) could influence serum GDF-15 concentrations. In addition, thyroid function, which regulates energy balance, can also influence both serum GDF-15 concentrations [20] and atherosclerosis as evaluated by CIMT [21]. Thyroid function also influences BMI [22]. To evaluate the association between serum GDF-15 concentrations and atherosclerosis as evaluated by CIMT, the influence of age, thyroid function, and BMI should be taken into consideration. We hypothesized that by indicating cellular energy balance, which is related to mitochondrial function, GDF-15 levels are positively associated with atherosclerosis as evaluated by CIMT among older individuals with normal weight and normal thyroid hormone levels whose systemic energy balance might be within the normal range.

To evaluate the association between serum GDF-15 concentration and atherosclerosis in a general population of older adults, a cross-sectional study of older Japanese individuals aged 60–69 years with normal weight (normal range of BMI) and normal thyroid hormone concentrations was conducted.

## 2. Materials and Methods

### 2.1. Study Population

Written consent forms were used to ensure that participants understood the objectives of the study when obtaining informed consent. This study was approved by the ethics committee of the Nagasaki University Graduate School of Biomedical Sciences (project registration number 14051404-13).

Since age could influence both CIMT [17] and GDF-15 [18], the analysis of the association between GDF-15 concentrations and atherosclerosis could be strongly confounded by age. To avoid the confounding influence of age, the study population should be within a narrow age range. However, from the biochemical perspective, age is an important factor that influences vascular health and mitochondrial function. Thus, individuals in an early stage of old age (60 to 69 years) are an appropriate target population for the present study, similar to our previous studies [23,24,25].

The study population comprised 816 Japanese individuals aged 60–69 years from Saza town in western Japan who underwent an annual medical examination in 2014. To avoid the influence of abnormal systemic energy balance, this study only included participants who were within the normal range of BMI and the normal range of free thyroxine (T4) and free triiodothyronine (T3) concentrations. Thus, participants without data on BMI (*n* = 1), BMI ≥ 25 kg/m^2^ (*n* = 187), or BMI < 18.0 kg/m^2^ (*n* = 35) were excluded.

The analysis among participants within the normal range of BMI could reduce the influence of abnormal nutritional status. Participants with a history of thyroid disease (*n* = 20), participants without data on thyroid hormone (*n* = 8) or thyroid stimulating hormone (TSH) concentrations (*n* = 2), and participants with abnormal thyroid hormone concentrations (*n* = 26) were also excluded. The normal range of thyroid hormone concentration was defined as free T3 concentrations of 2.1–4.1 pg/mL and free T4 concentrations of 1.0–1.7 ng/dL, as previously reported [26]. An abnormal thyroid gland could strongly affect cellular energy balance, which could confound the association between GDF-15 concentrations and atherosclerosis dramatically. Participants without data on smoking status (*n* = 1) were also excluded. The remaining 536 participants, with a mean age of 65.4 years (standard deviation (SD), 2.7 years; range, 60–69 years), were included in the study.

### 2.2. Data Collection and Laboratory Measurements

The methods used in the present study, including thyroid function evaluation, have been described elsewhere [21,27]. To measure GDF-15 concentrations, serum samples were diluted tenfold with specific diluents, as per the manufacturer’s instructions for the bead-based multiplexed immunoassay system with Luminex xMAP technology (Luminex Corporation, Austin, TX, USA) used. GDF-15 concentrations were measured. We also performed quality control, which confirmed that all values were within their expected ranges.

A trained interviewer obtained information on clinical characteristics such as history of thyroid disease, history of stroke, history of ischemic heart disease, smoking status, and drinking status. Body weight and height were measured using an automatic body composition analyzer (BF-220; Tanita, Tokyo, Japan). BMI (kg/m^2^) was calculated. Systolic blood pressure (SBP) was recorded at rest.

Fasting blood samples were collected. TSH, free T3, and free T4 levels were measured using standard procedures at the LSI Medience Corporation (Tokyo, Japan). Glycohemoglobin (HbA1c), triglycerides (TG), high-density lipoprotein cholesterol (HDLc), and creatinine levels were measured using standard procedures at SRL Inc. (Tokyo, Japan). The estimate glomerular filtration rate (eGFR) was calculated using an established method with three variables as proposed by a working group of the Japanese Chronic Kidney Disease Initiative [28].

Experienced vascular examiners measured the CIMT of the left and right common carotid arteries using LOGIQ Book XP with an 8L-RS linear ultrasound transducer (4.0–11.0 MHz) (GE Healthcare, Milwaukee, WI, USA). Maximum values for the left and right common CIMT were then calculated using semi-automated digital edge-detection software (Intimascope; MediaCross, Tokyo, Japan) via a previously described protocol [29]. The recently developed Intimascope software was used to increase the accuracy and reproducibility of CIMT values. This software semi-automatically recognizes the edges of the internal and external membranes of the artery and automatically determines the distance at a sub-pixel level, estimated to be 0.01 mm [30]. Atherosclerosis was diagnosed as CIMT > 1.1 mm because a normal CIMT value has been previously reported to be ≤1.1 mm [31].

The reproducibility and validity of CIMT values evaluated using the present technique were assessed in our previous study with men aged 36–79 years. The simple correlation coefficient (*p* value) was 0.91 (*p* < 0.001) for intra-observer agreement (*n* = 32) and 0.78 (*p* < 0.001) for interobserver agreement between two observers (*n* = 41) [32].

### 2.3. Statistical Analysis

Characteristics of the study population by GDF-15 tertile were expressed as means ± SD, except for male sex, smoking status, drinking status, medication use, physical activity, TSH, and TG. Male sex, smoking status, drinking status, anti-hypertensive medication use, glucose-lowering medication use, lipid-lowering medication use, physical activity (exercise), and physical activity (daily) were expressed as percentages. Since TSH and TG had skewed distributions, they were expressed as medians (interquartile range).

Physical activity (exercise) was based on responses to the following question: “Are you in a habit of doing exercise to sweat lightly for over 30 min a time, twice a week, for over a year?” Participants who answered “yes” were counted as having physical activity (exercise).

Physical activity (daily) was based on responses to the following question: “In your daily life, do you walk or do an equivalent amount of physical activity for more than 1 h a day?” Participants who answered “yes” were counted as having physical activity (daily).

To evaluate the statistical significance of participant characteristics in relation to serum GDF-15 tertiles, a trend test was performed. To evaluate the correlation between GDF-15 concentrations and variables directly associated with systemic energy balance, unadjusted and sex-adjusted partial correlation coefficients were calculated. To evaluate the correlation between CIMT and variables directly associated with systemic energy balance, unadjusted and sex-adjusted partial correlation coefficients were also calculated.

Aggressive endothelial repair, which increases CIMT, also induces insufficient endothelial repair, which leads to the progression of functional atherosclerosis as evaluated based on the cardio-ankle vascular index (CAVI), but not the progression of CIMT [33,34,35]. These studies indicated that continuous CIMT values are not appropriate for evaluating the influence of GDF-15 on vascular health. Thus, we defined atherosclerosis as CIMT > 1.1 mm [31] in the present study.

Logistic regression models were used to calculate odds ratios (ORs) and 95% confidence intervals (CIs) to determine the associations between serum GDF-15 concentration and atherosclerosis. Three different models were used to evaluate the association between GDF-15 and atherosclerosis. Model 1 adjusted only for age (years) and sex. Even among euthyroid individuals, thyroid function is associated with renal function [36]. Instead of adjusting for known cardiovascular risk factors, we created another model (Model 2) that adjusted for thyroid function and renal function. In Model 2, in addition to age and sex, we included potential confounding factors that directly influence GDF-15 concentration and CIMT (logarithmic value), such as TSH (logarithmic value), free T3, and eGFR. However, thyroid function is well known to be associated with classical cardiovascular risk factors such as hypertension [21], dyslipidemia [37], and obesity [22]. To evaluate the association between GDF-15 concentrations and atherosclerosis, there should be no adjustment for thyroid function in the model with cardiovascular risk factors. In Model 3, to adjust for known cardiovascular risk factors, we included SBP (mmHg), BMI (kg/m^2^), drinking status (none, often, daily), smoking status (never, former, current), TG (logarithmic value), HDLc (mg/dL), and HbA1c (%) in the model. To avoid the influence of current smoking, we also evaluated the association between GDF-15 concentrations and atherosclerosis among those who are not current smokers.

To validate the association between GDF-15 concentrations and atherosclerosis, the association between sex-specific GDF-15 tertiles and atherosclerosis was evaluated. To evaluate the association between continuous GDF-15 values and atherosclerosis, logarithmic values of GDF-15 were used.

All statistical analyses were performed with SAS for Windows (version 9.4; SAS Inc., Cary, NC, USA). All *p*-values for statistical tests were two-tailed, and *p*-values of <0.05 were regarded as statistically significant.

## 3. Results

### 3.1. Characteristics of the Study Population

Table 1 shows the characteristics of the entire study population. The percentage of men was 39.9% and the mean age was 65.2 ± 2.7 years.

Characteristics of the study population by GDF-15 tertile are shown in Table 2. Serum GDF-15 level was significantly positively associated with age and current smoker status and inversely associated with eGFR.

### 3.2. Correlation between Growth Differentiation Factor 15 (GDF-15), Carotid Intima-Media Thickness (CIMT), and Related Variables

Table 3 shows the correlation between GDF-15, CIMT, and related variables. GDF-15 was slightly but positively correlated with CIMT (simple correlation coefficient (r) = 0.09, *p* = 0.034) and age (r = 0.18, *p* < 0.001) and significantly inversely associated with eGFR (r = −0.23, *p* < 0.001). These correlations remained even after adjusting for sex. No significant correlations between GDF-15 and BMI (r = −0.04, *p* = 0.926) or between GDF-15 and TSH (r = 0.03, *p* = 0.477) were observed.

There was no correlation between GDF-15 and free T3 (r = −0.07, *p* = 0.103) in the crude model. After adjusting for sex, a weak inverse correlation was observed (r = −0.13, *p* = 0.002).

### 3.3. Association between GDF-15 Concentrations and Atherosclerosis

The associations between GDF-15 and atherosclerosis are shown in Table 4. A significant positive association between GDF-15 and atherosclerosis was observed. The sex- and age- adjusted OR and 95% CI for the logarithmic value of GDF-15 and atherosclerosis were 2.62 (1.67, 5.87). This significant association was unchanged even after adjusting for thyroid function and renal function (Model 2) and known cardiovascular risk factors (Model 3). The corresponding ORs and 95% CIs were 2.61 (1.15, 5.93) for Model 2 and 2.49 (1.08, 5.71) for Model 3, respectively.

### 3.4. Association between GDF-15 Concentrations and Atherosclerosis among Non-Current Smokers

The association between GDF-15 concentrations and atherosclerosis among non-current smokers is shown in Table 5. A significant positive association between GDF-15 concentrations and atherosclerosis was observed. The sex- and age- adjusted OR and 95% CI for the logarithmic value of GDF-15 and atherosclerosis were 3.04 (1.26, 7.34). This significant association was unchanged even after adjusting for thyroid function and renal function (Model 2) and known cardiovascular risk factors (Model 3). The corresponding ORs and 95% CIs were 3.22 (1.29, 8.02) for Model 2 and 2.73 (1.11, 6.73) for Model 3, respectively.

### 3.5. Sex-Specific Analysis

We also performed sex-specific analysis and found essentially the same associations for men and women. The age-adjusted ORs and 95% CIs for the logarithmic GDF-15 and atherosclerosis were 2.61 (0.88, 7.72) for men and 2.71 (0.80, 9.19) for women, respectively.

### 3.6. Analysis Stratified by Age Group

When the analysis was stratified by age group, essentially the same associations were found. The sex-adjusted partial correlation (*p* value) between age and CIMT and between age and GDF-15 concentration were 0.04 (*p* = 0.564) and 0.01 (*p* = 0.238) for participants aged < 65 years (*n* = 244), respectively. For those aged ≥ 65 years (*n* = 292), the corresponding values were 0.01 (*p* = 0.896) and 0.13 (*p* = 0.031), respectively. The sex- and age-adjusted ORs and 95% CI for atherosclerosis and logarithmic GDF-15 values were 5.36 (1.17, 25.54) for participants aged < 65 years and 2.04 (0.78, 5.28) for those aged ≥ 65 years.

## 4. Discussion

The major finding of the present study is that GDF-15 is positively associated with atherosclerosis as evaluated by CIMT among general older Japanese individuals with normal weight, independent of thyroid function and known cardiovascular risk factors.

In a previous study of 87 patients on maintenance hemodialysis and 45 sex- and age-matched healthy controls, a significant correlation between serum GDF15 concentration and CIMT was reported (r = 0.61, *p* < 0.001) [9]. Another previous study involving 60 patients with transfusion-dependent beta-thalassemia and 30 sex- and age-matched healthy controls showed that CIMT on both sides is statistically significant higher in cases (median, 0.08 cm) than in controls (median, 0.04 cm) [10].

In the present study, we found that serum GDF-15 concentration is significantly positively associated with atherosclerosis as evaluated by CIMT, even among older individuals with normal weight in the general population who have normal serum thyroid hormone concentrations.

The mechanism underlying the positive association between serum GDF-15 concentration and atherosclerosis among older individuals with normal weight has not been clarified yet. Although a positive correlation between current smoker status and GDF-15 concentrations was observed in the present study, current smoking might not explain the present results, because when the analysis was performed among non-current smokers, the association between serum GDF-15 concentrations and atherosclerosis became slightly stronger. 

Essentially the same associations were found between men and women. Sex might not have strongly influenced the present results.

Aging, BMI status, and thyroid hormone activity can influence both GDF-15 concentration and CIMT [17,18,19,20]. Those factors could act as determinants of the association between GDF-15 concentrations and atherosclerosis. However, to avoid the influence of these factors, our present study was conducted among participants in a narrow age range (60 to 69 years) who had normal BMI and thyroid hormone concentrations. No strong correlations between these factors and GDF-15 or CIMT were observed. Aging, BMI, and thyroid activity might not affect the association between GDF-15 concentrations and atherosclerosis as evaluated by CIMT.

Mitochondria control inflammatory responses [38]. Recently, the mitochondria, which play an important role in the production of cellular energy, were revealed to be involved in the differentiation and activation of immune cells such as monocytes, T cells, B cells, and macrophages [39].

Macrophages play an important role in the development of atherosclerotic lesions [40]. GDF-15 deficiency attenuates the activity of macrophages [13]. It is produced by activated macrophages [41,42]. Individuals with higher serum GDF-15 levels might have higher macrophage activity, indicating a higher risk of developing atherosclerosis.

GDF-15 has been recognized as a reliable biomarker of acute cardiovascular events in patients with acute coronary syndrome or stable coronary artery disease [43]. However, GDF-15 mediates anti-inflammatory effects in mice with acute myocardial infarction [15]. Physical conditions that increase serum GDF-15 levels might lead to the development of atherosclerosis, but GDF-15 might prevent atherosclerosis. Therefore, although a strong significant association between GDF-15 concentrations and atherosclerosis was observed in the present study, the correlation between GDF-15 concentrations and continuous values of CIMT was weak.

GDF-15 could promote vascular development in conjunction with vascular endothelial growth factor (VEGF) [44]. GDF-15 could act as a potent endothelial cell activator that promotes both normal and injury-related angiogenesis [45]. The presence of endothelial progenitor cells is required for the development of atherosclerosis as evaluated by CIMT [35,46]. The progression of atherosclerosis (greater CIMT) is also associated with the development of angiogenesis [24,34]. GDF-15 concentrations could be positively associated with atherosclerosis and an indicator of angiogenesis activity. Mitochondrial function might be associated with longitudinal growth velocity in children and adolescents [47]. Height might be inversely associated with inflammation and atherosclerosis among adult individuals with high BMI [48,49]. Since mitochondrial stress and dysfunction increase GDF-15 levels [2], GDF-15 could be positively associated with atherosclerosis.

This is the first study that revealed a positive association between serum GDF-15 concentration and atherosclerosis as evaluated by CIMT among older individuals with normal weight and thyroid hormone levels. To prevent cardiovascular disease, the Japanese government developed a strategy for reducing the number of individuals with metabolic syndrome and overnutrition [50]. However, in the present study, serum GDF-15 concentration was significantly associated with atherosclerosis among individuals with normal BMI. Although further investigation is necessary, the present results could help identify individuals at high risk for atherosclerotic disease for whom the Japanese government has not recommended preventing atherosclerotic diseases as part of health guidance.

Potential limitations of the present study warrant consideration. Although macrophages, VEGF, and endothelial progenitor cells might have important effects on the present results, we have no data about those variables. Although C-reactive protein (CRP) stimulates GDF-15 expression in endothelial cells [51], GDF-15 protects against the aging-mediated systemic inflammatory response [4]. Levels of CRP might affect the association between GDF-15 concentrations and atherosclerosis. A previous study reported that white blood cell (WBC) count and CRP levels each independently predict mortality among the oldest old [52]. Among the present study population, data on WBC count were available for 380 participants. The sex-, age- and WBC count-adjusted OR and 95% CI for the logarithmic value of GDF-15 and atherosclerosis was 2.70 (1.07, 6.77). Further investigation with information on those variables is necessary. The lack of available information on comorbidities might have influenced nutritional status, which might have influenced the present results. However, the present analysis included older participants in the general population with BMI within the normal range, which might have reduced the influence of abnormal nutritional status. In addition, this was a cross-sectional study, which cannot establish causal relationships.

## 5. Conclusions

In conclusion, among older individuals with normal weight and normal thyroid hormone levels, the serum GDF-15 concentration was revealed to be significantly positively associated with atherosclerosis. Serum GDF-15 concentration could be an efficient tool for estimating the risk of atherosclerosis among older individuals with normal weight. Serum GDF-15 concentration might reflect factors associated with cellular energy balance that might lead to the progression of atherosclerosis, even among older individuals with normal weight and normal thyroid hormone levels. The present results could help identify individuals at high risk for atherosclerotic disease for whom the Japanese government has not recommended atherosclerotic disease prevention as part of health guidance.

## Figures and Tables

**Table 1 biomedicines-11-01572-t001:** Characteristics of the study population.

	Total
No. of participants	536
Men, %	39.9
Age	65.2 ± 2.7
History of stroke, %	3.4
History of ischemic heart disease, %	5.0
TSH, (0.39–4.01) μIU/mL	1.50 [1.10, 2.32] *^1^
free T3, (2.1–4.1) pg/mL	3.2 ± 0.3
free T4, (1.0–1.7) ng/dL	1.2 ± 0.2
Body mass index, kg/m^2^	21.8 ± 1.9
Systolic blood pressure, mmHg	127 ± 15
Drinker (Daily), %	39.9
Drinker, (Often) %	2.4
Current smoker, %	13.4
Former smoker, %	21.3
Antihypertensive medication use, %	32.6
Glucose lowering medication use, %	5.2
Lipid lowering medication use, %	25.9
Physical activity (exercise), %	42.0
Physical activity (daily), %	91.0
Triglycerides, mg/dL	88 [67, 122] *^1^
HDLc, mg/dL	61 ± 15
HbA1c, %	5.6 ± 0.6
eGFR, mL/min/1.73 m^2^	70.0 ± 12.4
CIMT, mm	0.84 [0.75, 0.97] *^1^

TSH: thyroid stimulating hormone; T3: triiodothyronine; T4: thyroxine; TG: triglycerides; HDLc: HDL-cholesterol; HbA1c: glycohemoglobin; eGFR: estimate glomerular filtration rate. Values are mean ± standard deviation. *^1^: Values are median [the first quartile, the third quartile].

**Table 2 biomedicines-11-01572-t002:** Characteristics of the study population in relation to serum growth differentiation factor 15 (GDF-15) levels.

	GDF-15			*p*
	T1 (Low)	T2 (Middle)	T3 (High)	
No. of participants	177	183	176	
Men, %	39.0	40.4	40.3	0.952
Age	64.4 ± 2.8	65.3 ± 2.6	65.7 ± 2.6	<0.001
History of stroke, %	2.8	1.6	5.7	0.093
History of ischemic heart disease, %	2.8	4.9	7.4	0.147
TSH, (0.39–4.01) μIU/mL	1.49 [1.06, 2.28] *^1^	1.59 [1.15, 2.34] *^1^	1.45 [1.09, 2.30] *^1^	0.385 *^2^
free T3, (2.1–4.1) pg/mL	3.2 ± 0.3	3.2 ± 0.3	3.2 ± 0.3	0.159
free T4, (1.0–1.7) ng/dL	1.2 ± 0.2	1.2 ± 0.2	1.2 ± 0.2	0.910
Body mass index, kg/m^2^	21.9 ± 1.9	21.8 ± 1.7	21.9 ± 1.9	0.891
Systolic blood pressure, mmHg	127 ± 15	127 ± 16	128 ± 15	0.637
Drinker (Daily), %	42.4	42.1	35.2	0.300
Drinker, (Often) %	1.7	1.6	4.0	0.265
Current smoker, %	6.2	18.2	19.9	<0.001
Former smoker, %	23.2	19.1	21.6	0.641
Antihypertensive medication use, %	27.7	31.7	38.6	0.085
Glucose lowering medication use, %	2.8	6.0	6.8	0.204
Lipid lowering medication use, %	26.0	25.7	26.4	0.995
Physical activity (exercise), %	42.9	43.2	39.8	0.770
Physical activity (daily), %	90.4	91.2	90.9	0.894
Triglycerides, mg/dL	85 [63, 119] *^1^	92 [67, 129] *^1^	88 [67, 118] *^1^	0.463 *^2^
HDLc, mg/dL	61 ± 14	61 ± 15	60 ± 15	0.827
HbA1c, %	5.6 ± 0.4	5.6 ± 0.6	5.7 ± 0.6	0.146
eGFR, mL/min/1.73 m^2^	72.9 ± 11.0	68.7 ± 11.1	68.3 ± 14.4	<0.001
CIMT, mm	0.84 [0.75, 0.95] *^1^	0.83 [0.77, 0.96] *^1^	0.85 [0.75, 0.99] *^1^	0.056 *^2^

TSH: thyroid stimulating hormone; T3: triiodothyronine; T4: thyroxine; TG: triglycerides, HDLc: HDL-cholesterol; HbA1c: glycohemoglobin; eGFR: estimate glomerular filtration rate. Values are mean ± standard deviation. *^1^: Values are the median [first quartile, third quartile]. *^2^: Logarithmic transformation was used for evaluating p. Tertile values of GDF-15 for men were <0.82 ng/mL for T1 (low), 0.82–1.07 ng/mL for T2 (middle), and 1.08 ng/mL≤ for T3 (high). The corresponding values for women were <0.68 ng/mL, 0.68–0.90 ng/mL, and 0.91 ng/mL≤.

**Table 3 biomedicines-11-01572-t003:** Correlation between serum growth differentiation factor 15 (GDF-15), carotid intima-media thickness (CIMT) and variables.

		**GDF-15**		**CIMT *^1^**		**Age**		**BMI**	
		**r**	** *p* **	**r**	** *p* **	**r**	** *p* **	**r**	** *p* **
GDF-15	Simple	-	-	0.09	0.034	0.18	<0.001	−0.04	0.926
	Partial	-	-	0.08	0.074	0.18	<0.001	−0.04	0.317
CIMT *^1^	Simple	0.09	0.034	-	-	0.13	0.002	0.08	0.075
	Partial	0.08	0.074	-	-	0.13	0.002	0.07	0.119
		**TSH *^1^**		**free T3**		**free T4**		**eGFR**	
		**r**	** *p* **	**r**	** *p* **	**r**	** *p* **	**r**	** *p* **
GDF-15	Simple	0.03	0.477	−0.07	0.103	0.04	0.335	−0.23	<0.001
	Partial	0.05	0.211	−0.13	0.002	0.003	0.945	−0.25	<0.001
CIMT *^1^	Simple	0.02	0.572	−0.04	0.394	−0.03	0.625	−0.01	0.891
	Partial	0.03	0.481	−0.05	0.214	−0.03	0.455	−0.01	0.833

*^1^: Logarithmic transformation was used. Partial: adjusted for sex. BMI: body mass index. TSH: thyroid stimulating hormone. T3: triiodothyronine. T4: thyroxine. eGFR: estimate glomerular filtration rate.

**Table 4 biomedicines-11-01572-t004:** Association between growth differentiation factor 15 (GDF-15) and atherosclerosis.

	GDF-15			*p* for Trend	GDF-15 (Logarithmic Values)
	T1 (Low)	T2 (Middle)	T3 (High)		
No. of participants	177	183	176		
No. of cases (%)	14 (7.9)	25 (13.7)	29 (16.5)		
Model 1	Referent	1.57 (0.78, 3.18)	2.08 (1.05, 4.14)	0.036	2.62 (1.67, 5.87)
Model 2	Referent	1.59 (0.78, 3.24)	2.08 (1.04, 4.17)	0.039	2.61 (1.15, 5.93)
Model 3	Referent	1.53 (0.75, 3.15)	1.99 (0.97, 4.06)	0.060	2.49 (1.08, 5.71)

Model 1: adjusted only for sex and age. Model 2: further adjusted (Model 1 +) for thyroid stimulating hormone (TSH), free triiodothyronine (T3), and estimate glomerular filtration rate (eGFR). Model 3: further adjusted (Model 1 +) for systolic blood pressure (SBP), body mass index (BMI), drinking status, smoking status, triglycerides (TG), HDL-cholesterol (HDLc), glycohemoglobin (HbA1c). Tertile values of GDF-15 for men were <0.82 ng/mL for T1 (low), 0.82–1.07 ng/mL for T2 (middle), and 1.08 ng/mL≤ for T3 (high). The corresponding values for women were <0.68 ng/mL, 0.68–0.90 ng/mL, and 0.91 ng/mL≤, respectively.

**Table 5 biomedicines-11-01572-t005:** Association between growth differentiation factor 15 (GDF-15) and atherosclerosis among non-current smokers.

	GDF-15			*p* for Trend	GDF-15 (Logarithmic Value)
	T1 (Low)	T2 (Middle)	T3 (High)		
No. of participants	166	157	141		
No. of cases (%)	12 (7.2)	18 (11.5)	25 (17.7)		
Model 1	Referent	1.58 (0.73, 3.42)	2.60 (1.23, 5.49)	0.011	3.04 (1.26, 7.34)
Model 2	Referent	1.60 (0.73, 3.48)	2.65 (1.24, 5.66)	0.011	3.22 (1.29, 8.02)
Model 3	Referent	1.56 (0.71, 3.43)	2.48 (1.15, 5.33)	0.019	2.73 (1.11, 6.73)

Model 1 only adjusted for sex and age. Model 2 adjusted for variables in Model 1 as well as thyroid stimulating hormone (TSH), free triiodothyronine (T3), and estimated glomerular filtration rate (eGFR). Model 3 adjusted for variables in Model 1 as well as systolic blood pressure (SBP), body mass index (BMI), drinking status, smoking status (never or former), triglycerides (TG), HDL-cholesterol (HDLc), and glycohemoglobin (HbA1c). GDF tertiles for men were <0.82 ng/mL for T1 (low), 0.82–1.07 ng/mL for T2 (middle), and 1.08 ng/mL≤ for T3 (high). The corresponding values for women were <0.68 [ng/mL], 0.68–0.90 ng/mL, and 0.91 ng/mL≤, respectively.

## Data Availability

We cannot publicly provide individual data due to participant privacy, according to ethical guidelines in Japan. Additionally, the informed consent was obtained does not include a provision for publicity sharing data. Qualifying researchers may apply to access a minimal dataset by contacting Prof Naomi Hayashida, Principal Investigator, Division of Promotion of Collaborative Research on Radiation and Environment Health Effects, Atomic Bomb Disease Institute, Nagasaki University, Nagasaki, Japan at naomin@nagasaki-u.ac.jp. Or, please contact the office of data management at ritouken@vc.fctv-net.jp. Information for where data request is also available at https://www.genken.nagasaki-u.ac.jp/dscr/message/ (accessed on 23 May 2023) and http://www.med.nagasaki-u.ac.jp/cm/ (accessed on 23 May 2023).

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
