# Peer review of "Serum Concentration of Growth Differentiation Factor 15 and Atherosclerosis among General Older Japanese Individuals with Normal Weight"

_biomedicines, 2023, doi:10.3390/biomedicines11061572_

Round 1

Reviewer 1 Report

This study analyzed the pure effect of GDF-15 serum concentration and atherosclerosis. This study is carefully performed and of interest.

The main problem is that it is difficult to follow the authors on their way to conclusions.  At the end I was no convinced that the conclusion is realistic.

The major finding of this study is that GDF-15 is positively associated with atherosclerosis. It was really difficult to understand how the authors come to this conclusion. Obviously, it is based on Table 4, defining atherosclerosis as CIMT >1.1. Even if we accept this it is noteworthy that significance of this effect is lost in Model 3. I do not want to strength the p-value that is a little bit above p=0.05. However, given the strong correlation between smoking and GDF-15 (Table 2) and the absence of a strong correlation between serum GDF-15 and CIMT (Table 2) it seems that smoking and not GDF-15 drives CIMT and atherosclerosis. As smoking affects energy turnover and GDF-15 is linked to energy metabolism this is not a surprise. However, it seems that valuating the smoking status is more important than to estimate GDF-15 serum levels.

The second problem is the correlations shown in Table 3. It is rather unbelievable that that an r value as 0.09 results in significant effects. It would really help to see the original correlation graphs.

Finally, the study could be improved if the authors can calculate cutoff values for GDF-15.

In the discussion the authors debate about macrophages, inflammation, and GDF-15. DO you have any data on immune cell counts? This would strengthen the mechanistic speculation.

Author Response

This study analyzed the pure effect of GDF-15 serum concentration and atherosclerosis. This study is carefully performed and of interest.

(Comment 1)

The main problem is that it is difficult to follow the authors on their way to conclusions.  At the end I was no convinced that the conclusion is realistic.

Thank you for valuable comment. According to this reviewer’s valuable comment, I re-checked the concept of present study. GDF-15 is known factor that positively associated with mitochondrial dysfunction. And because mitochondrial takes important role in production of cellular energy, lower producibility of cell energy might underlying the positive association between GDF-15 and cardiovascular diseases which is reported by previous studies.

However, among older, abnormal BMI and abnormal thyroid function is well known causes of abnormal energy balance. Then the finding that was shown in present study (positive association between GDF-15 and atherosclerosis among older with normal thyroid hormone levels and normal BMI) indicates that even among those with normal cellar energy balance, mitochondrial activity might influence on development of atherosclerosis. This is what we thought interesting.

Then we revised as following in the conclusion of abstract.

Serum GDF-15 concentrations could help estimate the risk of atherosclerosis by indicating the status of cellular energy balance, which is related to mitochondrial activity among comparative healthy older individuals.

In addition to that, in discussion section we discussed about the strategy for prevent cardiovascular diseases that was developed by the Japanese government.

To prevent cardiovascular disease, the Japanese government developed a strategy for reducing the number of individuals with metabolic syndrome and overnutrition [49]. However, in the present study, serum GDF-15 concentration was significantly associated with atherosclerosis among individuals with normal BMI. Although further investigation is necessary, the present results could help identify individuals at high risk for atherosclerotic disease for whom the Japanese government has not recommended preventing atherosclerotic diseases as part of health guidance.

And after explaining the strategy to prevent atherosclerotic disease which is performed by Japanese government, we made conclusion as following.

The present results could help identify individuals at high risk for atherosclerotic disease for whom the Japanese government has not recommended atherosclerotic disease prevention as part of health guidance.

(Comment 2-1)

The major finding of this study is that GDF-15 is positively associated with atherosclerosis. It was really difficult to understand how the authors come to this conclusion. Obviously, it is based on Table 4, defining atherosclerosis as CIMT >1.1. Even if we accept this it is noteworthy that significance of this effect is lost in Model 3. I do not want to strength the p-value that is a little bit above p=0.05.

Thank you for valuable comment. According to this reviewer’s valuable comment, I re-checked the result of the Table 4.

As is often the case of epidemiological study, to evaluate the association between serum GDF15 and atherosclerosis among present study population, we performed the analysis by using mainly two separate model.

In this study, the first one evaluates the tertile values of serum GDF-15 and atherosclerosis. In this analysis, we evaluate the tendency of the association between serum GDF15 and atherosclerosis. Then, instead of showing the significant values of p value, the results that showed step up association between the tertile values of serum GDF-15 and atherosclerosis is important. Even if significant association between tertile of GDF-15 and atherosclerosis was observed in this model, that result only indicates that the classification shows enough value to reach significant level that is not main purpose of this type of analysis.

And the next one is calculating the association between logarithmic values of GDF15 and atherosclerosis which is intended to evaluate the continuous values of GDF-15 and atherosclerosis. And we found significant value. This indicates that there is a significant association between the continuous values of GDF-15 and atherosclerosis.

Then, only by evaluating those two models, we could conclude that there is significant positive association between those two.

If the association between tertiles of GDF15 and atherosclerosis showed U shape but significant values were observed between logarithmic values of GDF15 and atherosclerosis, we could not conclude that we have significant association between those two.

This is the reason why we could conclude that there is significant positive association between GDF15 and atherosclerosis only by using those two different types of analyzes.

(Comment 2-2)

However, given the strong correlation between smoking and GDF-15 (Table 2) and the absence of a strong correlation between serum GDF-15 and CIMT (Table 2) it seems that smoking and not GDF-15 drives CIMT and atherosclerosis. As smoking affects energy turnover and GDF-15 is linked to energy metabolism this is not a surprise. However, it seems that valuating the smoking status is more important than to estimate GDF-15 serum levels.

Thank you for valuable comment. According to this reviewer’s valuable comment, I re-checked the influence of current smoker. And we made main analysis limited to non-current smoker.

Then in statistical analysis section we added following sentence.

To avoid the influence of current smoking, we also evaluated the association between GDF-15 concentrations and atherosclerosis among those who are not current smokers.

And in result section we added Table 5 and we also added following sentences in result section.

3.3. Association between GDF-15 concentrations and atherosclerosis     among non-current smokers

The association between GDF-15 concentrations and atherosclerosis among non-current smokers is shown in Table 5. A significant positive association between GDF-15 concentrations and atherosclerosis was observed. The sex- and age- adjusted OR and 95% CI for the logarithmic value of GDF-15 and atherosclerosis was 3.04 (1.26, 7.34). This significant association was unchanged even after adjusting for thyroid function and renal function (Model 2) and known cardiovascular risk factors (Model 3). The corresponding ORs and 95% CIs were 3.22 (1.29, 8.02) for Model 2 and 1.11 (1.11, 6.73) for Model 3, respectively.

      Table 5. Association between growth differentiation factor 15 (GDF-15) concentrations and atherosclerosis among non-current smoker

GDF-15

p for trend

GDF-15

(Logarithmic value)

T1 (low)

T2 (middle)

T3 (high)

No. at risk 

166

157

141

No. of participants (%)

12 (7.2)

18 (11.5)

25 (17.7)

 Model 1

Referent

1.58

(0.73, 3.42)

2.60

(1.23, 5.49)

0.011

3.04

(1.26, 7.34)

 Model 2

Referent

1.60

(0.73, 3.48)

2.65

(1.24, 5.66)

0.011

3.22

(1.29, 8.02)

 Model 3

Referent

1.56

(0.71, 3.43)

2.48

(1.15, 5.33)

0.019

2.73

(1.11, 6.73)

Model 1 only adjusted for sex and age. Model 2 adjusted for variables in Model 1 as well as thyroid stimulating hormone (TSH), free triiodothyronine (T3), and estimated glomerular filtration rate (eGFR). Model 3 adjusted for variables in Model 1 as well as systolic blood pressure (SBP), body mass index (BMI), drinking status, smoking status (never or former), triglycerides (TG), HDL-cholesterol (HDLc), and glycohemoglobin (HbA1c). GDF tertiles for men were < 0.82 ng/mL for T1 (low), 0.82-1.07 ng/mL for T2 (middle), and ≥1.07 ng/mL for T3 (high). The corresponding values for women were <0.68 [ng/mL], 0.68-0.90 ng/mL, and ≥0.91 ng/mL.

(Comment 2-3)

The second problem is the correlations shown in Table 3. It is rather unbelievable that that an r value as 0.09 results in significant effects. It would really help to see the original correlation graphs.

Thank you for valuable comment. According to this reviewer’s valuable comment, I re-checked the impact of GDF-15 on atherosclerosis. And we revised as following in discussion section.

GDF-15 has been recognized as a reliable biomarker of acute cardiovascular events in patients with acute coronary syndrome or stable coronary artery disease [43]. However, GDF-15 mediates anti-inflammatory effects in mice with acute myocardial infarction [15]. Physical conditions that increase serum GDF-15 levels might lead to the development of atherosclerosis but GDF-15 might prevent atherosclerosis. Therefore, although a strong significant association between GDF-15 concentrations and atherosclerosis was observed in the present study, the correlation between GDF-15 concentrations and continuous values of CIMT was weak.

(Comment 3)

Finally, the study could be improved if the authors can calculate cutoff values for GDF-15.

Thank you for valuable comment. According to this reviewer’s valuable comment, we made ROC curve by using the logarithmic values of GDF15 and atherosclerosis in fully adjusted model, area under curve shows 0.67, the association was used as logarithmic value. Then influence of misclassification could be strong in present study.

(Comment 4)

In the discussion the authors debate about macrophages, inflammation, and GDF-15. DO you have any data on immune cell counts? This would strengthen the mechanistic speculation.

Thank you for valuable comment. According to this reviewer’s valuable comment, we rechecked the influence of inflammatory marker on present study. And we added following sentences in limitation section.

A previous study reported that white blood cell (WBC) count and CRP levels each independently predict mortality among the oldest old [51]. Among the present study population, data on WBC count was available for 380 participants. The sex-, age- and WBC count-adjusted OR and 95% CI for the logarithmic value of GDF-15 and atherosclerosis was 2.70 (1.07, 6.77). Further investigation with information on those variables is necessary.

Reviewer 2 Report

Very interesting manuscript. It is still necessary to search for markers of prognostic significance, as well as those influencing potential therapeutic procedures. Work presented correctly, well discussed.
I believe that the authors focused very much on the problem of thyroid function (which is understandable due to the "energetics"), but it would be advisable to clarify additional elements of the description of the population:

- please correct the wording about the exclusion criteria. the sentence "to reduce the influence of severe degenerative diseases such as
Inflammatory disease, chronic obstructive pulmonary disease, cancer, and cardiovascular disease, participants with low BMI were excluded." It continues "To reduce the influence of severe heart 104
failure, individuals with high BMI were also excluded". BMI is not used to diagnose the listed diseases, so it is only true that the authors excluded people with a BMI lower or higher than the norm.
- have the persons been treated, received drugs that may affect the function of the vascular endothelium? smooth muscle.
- how was the physical activity of these people? it also affects the function of the vascular endothelium.
- whether there were people with high physical activity in the population. There have been published manuscripts highlighting the differences in physical activity - aging athlete's heart: an echocardiographic evaluation of competitive sprint - versus endurance-trained master athletes. Was this not the case in the study group? this could have a significant influence on the observed results.
- were there any diseases that could significantly modulate the activity of the vegetative system?

- please also extend the conclusion. The conclusion should also summarize the practical aspects and perspectives resulting from the conducted research.

Author Response

(Comment. 1)
- please correct the wording about the exclusion criteria. the sentence "to reduce the influence of severe degenerative diseases such as
Inflammatory disease, chronic obstructive pulmonary disease, cancer, and cardiovascular disease, participants with low BMI were excluded." It continues "To reduce the influence of severe heart failure, individuals with high BMI were also excluded". BMI is not used to diagnose the listed diseases, so it is only true that the authors excluded people with a BMI lower or higher than the norm.

Thank you for valuable comment. According to this reviewer’s valuable comment, we deleted the following sentences.

To reduce the influence of severe degenerative diseases such as inflammatory disease, chronic obstructive pulmonary disease, cancer, and cardiovascular disease, participants with low BMI were excluded. To reduce the influence of severe heart failure, individuals with high BMI were also excluded from the present study.

(Comment 2)
- have the persons been treated, received drugs that may affect the function of the vascular endothelium? smooth muscle.

Thank you for valuable comment. According to this reviewer’s valuable comment, we added the percentages of taking medication for hypertension, diabetes, and hyperlipidemia in table 1 and table2. As shown in table 2, no of those factors revealed to be associated with GDF15 levels and in adjusted model we already adjusted for systolic blood pressure, triglycerides, HDL-cholesterol, and HbA1c, further adjustment for those factors might evoke the risk of multi-covariance. Then we did not include those factors as confounder in present model.

(Comment 3)

- how was the physical activity of these people? it also affects the function of the vascular endothelium.

Thank you for valuable comment. According to this reviewer’s valuable comment, we recheck the questionnaire which was performed in present health check-up. And we have the data of physical activity (exercise) and physical activity (daily) in present. Then we added those data in table 1 and table 2.

And we also added the following sentences in method section.

Characteristics of the study population by GDF-15 tertile were expressed as means ± SD, except for male sex, smoking status, drinking status, TSH, and TG. Male sex, smoking status, anti-hypertensive medication use, glucose-lowering medication use, lipid-lowering medication use, drinking status, physical activity (exercise), and physical activity (daily) were expressed as percentages. Since TSH and TG had skewed distributions, they were expressed as medians (interquartile range).

Physical activity (exercise) status was based on responses to the following question: “Are you in a habit of doing exercise to sweat lightly for over 30 minutes a time, twice a week, for over a year?” Participants who answered “yes” were counted as having physical activity (exercise).

Physical activity (daily) was based on responses to the following question: “In your daily life, do you walk or do an equivalent amount of physical activity for more than 1 hour a day?” Participants who answered “yes” were counted as having physical activity (daily).

(Comment 4)

- whether there were people with high physical activity in the population. There have been published manuscripts highlighting the differences in physical activity - aging athlete's heart: an echocardiographic evaluation of competitive sprint - versus endurance-trained master athletes. Was this not the case in the study group? this could have a significant influence on the observed results.

Thank you for valuable comment. According to this reviewer’s valuable comment, we recheck the study population of present study. Since this study is performed among general older population, the influence of aging athlete’s heart should be limited. And we have no associations between serum GDF15 levels and physical activity (exercise), and the association between GDF15 and physical activity (daily). Then we did not added those factors as confounder in present model.

(Comment 5)
- were there any diseases that could significantly modulate the activity of the vegetative system?

Thank you for valuable comment. According to this reviewer’s valuable comment, we reconsider what disease could influence on present associations. Since this is an observational study that was performed among general population, rare disease should not influence on present results. Since overt and subclinical hypothyroidism is often observed among general older population, thyroid function should be checked. And for other diseases the abnormal nutritional status should be second causes of those disease. Then analysis limited to those with normal BMI could reduce the influence of severe degenerative disease. This is the part of the reason why we evaluate the association between GDF15 and atherosclerosis among older with normal range of BMI and normal thyroid hormone. Then we added following sentences in limitation section.

Lack of available information on comorbidities might have influenced nutritional status, which might have influenced the present results. However, the present analysis included older participants in the general population with BMI within the normal range, which might have reduced the influence of abnormal nutritional status. In addition, this was a cr

(Comment 6)

- please also extend the conclusion. The conclusion should also summarize the practical aspects and perspectives resulting from the conducted research.

Thank you for valuable comment. According to this reviewer’s valuable comment, we added following sentences in conclusion section.

The present results could help identify individuals at high risk for atherosclerotic disease for whom the Japanese government has not recommended atherosclerotic disease prevention as part of health guidance.

Round 2

Reviewer 1 Report

I thank the authors for re-submission of the manuscript and responding to my questions. Most points could be clarified. Concerning smoking I am still not convinced that you came to the right conclusions. If smoking affects GDF15 it clear that you can mathematicval correct come to your association. However, as smoking affects energy metabolism and GDF15 eneergy metabolism this is not a surprise. It seems for me that you can come to the same conclusion (low BMI but high risk of atherosclerosis) if you replace GDF15 by smoking. Smoking would give a machnistic link to atherosclerosis but not GDF15. Therefore, I am not convinced that GDF15 quantification adds anything to the risk stratification that excceds the level of smoking.

Author Response

I thank the authors for re-submission of the manuscript and responding to my questions. Most points could be clarified. Concerning smoking I am still not convinced that you came to the right conclusions. If smoking affects GDF15 it clear that you can mathematicval correct come to your association. However, as smoking affects energy metabolism and GDF15 eneergy metabolism this is not a surprise. It seems for me that you can come to the same conclusion (low BMI but high risk of atherosclerosis) if you replace GDF15 by smoking. Smoking would give a machnistic link to atherosclerosis but not GDF15. Therefore, I am not convinced that GDF15 quantification adds anything to the risk stratification that excceds the level of smoking.

Thank you for your comment. According to this reviewer’s valuable comment, I rechecked the influence of smoking on present associations.

Even GDF-15 is revealed to be positively associated with current smoker, while smoking is well known risk of development of atherosclerosis, smoking never mediate present associations because of the following reasons.

First, the prevalence of smoker is much higher in men than that of women. However, our sensitivity analysis revealed that essentially same associations were observed between men and women. Then we did describe as following in result section.

3.4. Sex-specific analysis

We also performed sex-specific analysis and found essentially same associations for men and women. The age-adjusted ORs and 95% CIs for the logarithmic GDF-15 and atherosclerosis were 2.61 (0.88, 7.72) for men and 2.71 (0.80, 9.19) for women, respectively.

And second, in present study when we made further analysis among non-smoker (non-current smoker), the association between GDF-15 and atherosclerosis became slightly stronger. As shown in table 1, current smoker but not former smoker shows significantly positively associated with GDF-15. Then ,this indicates that after removing the influence of smoking, the association between GDF-15 and atherosclerosis became slightly stronger. Then smoking act as a confounder on present association but not mediator.

If smoking mediates the association between GDF-15 and atherosclerosis, the statistical value became non-significant level when the analysis limited to non-smoker (non-current smoker). This is the reason why we did describe as following in discussion section.

Although a positive correlation between current smoker status and GDF-15 concentrations was observed in the present study, current smoking might not explain the present results because when the analysis was performed among non-current smokers, the association between serum GDF-15 concentrations and atherosclerosis became slightly stronger.

And the third, because we already made the analysis non-smoker (non-current smoker), GDF-15 value could not replace by smoking status. This also explains that independent from smoking status, GDF-15 found to be significantly positively associated with atherosclerosis.

And last, in present study we made 3 models when adjusting for confounder. And only in model 3, the status of smoking took as a confounder. If the influence of smoking on the association between GDF-15 and atherosclerosis is strong, after adjustment for smoking status the association became significantly weak. However, essentially same magnitude of the association between GDF-15 and atherosclerosis were observed among those models (Model 1, Model 2, and Model 3). Then the influence of smoking on the association between GDF-15 and atherosclerosis revealed to be weak.

Therefore, we could conclude that the status of smoking never explains the association between GDF-15 and atherosclerosis in present study.

Reviewer 2 Report

I believe that the revised manuscript may be considered for publication

Author Response

I believe that the revised manuscript may be considered for publication

Thank you for your comment.